# Decrease in Socioeconomic Disparities in Self-Rated Oral Health among Brazilian Adults between 2013 and 2019: Results from the National Health Survey

**DOI:** 10.3390/ijerph21091198

**Published:** 2024-09-10

**Authors:** Anna Rachel dos Santos Soares, Maria Luiza Viana Fonseca, Deborah Carvalho Malta, Loliza Luiz Figueiredo Houri Chalub, Raquel Conceição Ferreira

**Affiliations:** 1Department of Social and Community Dentistry, School of Dentistry, Universidade Federal de Minas Gerais, Belo Horizonte 31270-901, MG, Brazil; annasoares@ufmg.br (A.R.d.S.S.); mlvfonseca@ufmg.br (M.L.V.F.); lcfigueiredo@ufmg.br (L.L.F.H.C.); 2Department of Maternal and Child Nursing and Public Health, School of Nursing, Universidade Federal de Minas Gerais, Belo Horizonte 30130-100, MG, Brazil; dcmalta@uol.com.br

**Keywords:** oral health, self-concept, adults, health inequality monitoring, health status disparities, dental health surveys

## Abstract

This cross-sectional study assessed the magnitude of inequalities in self-rated oral health (SROH) among different socioeconomic groups in Brazil. Secondary data from interviews with a sample of adults (≥18 years) from the national health survey 2013 (*n* = 64,308) and 2019 (*n* = 88,531) were analyzed. Positive SROH was considered when participants selected the good or very good options. Socioeconomic indicators were monthly household income and years of education. The magnitude of inequalities among socioeconomic groups was estimated using the Slope (SII) and Relative Index of Inequality (RII). Interaction term assessed changes in SII/RII over time. Estimates were adjusted for sex and age. The prevalence of SROH was 67.50% in 2013 and 69.68% in 2019. Individuals with lower socioeconomic indicators had a lower prevalence of positive SROH. Significant reductions in the magnitude of the education-based RII between 2013 (1.58) and 2019 (1.48) in Brazil, as well as in north (1.70; 1.45) and northeast (1.50; 1.41) regions and reduction in the income-based RII in the north (1.71; 1.51) were observed. Socioeconomic inequalities in SROH persist across different Brazilian regions, although there was a reduction in disparities among education groups in 2019 compared with 2013. The findings of this study suggest that equitable Brazilian oral health policies may have contributed to reducing SROH inequality over time.

## 1. Introduction

Self-rated oral health (SROH) refers to individuals’ perceptions of their satisfaction and self-esteem with oral health related to comfort while eating, sleeping, and interacting socially. It provides a comprehensive and sensitive measure of health status, encompassing physical and psychosocial dimensions within individual cultural and environmental contexts [1]. Subjective measures offer valuable insights into overall health, surpassing the limitations of objective indicators [2] as they evaluate an individual’s physical and psychosocial health, social well-being, and quality of life. This evaluation is crucial because an oral problem affects an individual’s daily life only if it is perceived [3]. Positive SROH has been associated with favorable socioeconomic conditions, whereas negative SROH has been linked to unfavorable socioeconomic and demographic factors, following a social gradient of health inequalities [1,4]. 

Socioeconomic factors have been associated with oral health problems by affecting dietary habits, oral hygiene practices, and access to dental services [4]. Both education and income are robustly associated with health status [2,5]. Limited socioeconomic resources and precarious living conditions create barriers to dental services, resulting in inadequate oral health care, oral disease, tooth loss, and compromised SROH [3,4,6]. In contrast, higher education and income levels are associated with better perceived overall health status [5,7].

In Brazil, the Ministry of Health launched the National Oral Health Policy (NOHP) in 2004 to reduce inequalities in oral health. This policy has expanded dental health coverage within primary healthcare settings alongside initiatives such as water fluoridation, integration of primary care through the Family Health Program, and the establishment of specialized centers. Despite the integration of oral health into the Unified Health System (SUS) in Brazil, significant oral health issues persist, particularly among those socioeconomically vulnerable individuals [8] with persistent untreated oral conditions, including untreated caries, severe periodontitis, total tooth loss [9,10], and higher prevalence of negative SROH [4]. Therefore, monitoring disparities within the NOHP framework is crucial for evaluating how policies reach vulnerable populations reliant on public health services. Additionally, understanding and describing oral health disparities can guide the restructuring of dental services to better meet the needs of minority groups affected by oral health impacts on their daily lives [4]. This study represents a pioneering effort to quantify the relative and absolute magnitude of socioeconomic disparities in Brazil, utilizing robust methodologies consistent with the standards set by the World Health Organization [5]. Furthermore, it evaluates changes in these disparities by comparing findings from two national epidemiological surveys, which were previously estimated through absolute differences in the prevalence of SROH among social groups, without considering the population distribution of the groups [10].

Despite the significant progress made, there is still a lack of comprehensive studies that evaluate the long-term effect of the NOHP in reducing oral health inequalities among different socioeconomic and regional groups in Brazil. This study addresses this gap by providing a detailed analysis of how disparities in oral health have evolved. Then, this study aims to monitor disparities in oral health about a decade after the implementation of the NOHP, with a second evaluation conducted six years later. Recognizing the vast geographical, economic, and cultural diversity in the country, the objective was to investigate the magnitude of disparities in positive SROH among socioeconomic groups in Brazil between 2013 and 2019 and across Brazilian regions. Our hypothesis is that socioeconomic disparities decreased between 2013 and 2019.

## 2. Materials and Methods

This analytical cross-sectional study used public secondary data from two national health surveys (NHSs) conducted in Brazil in 2013 and 2019. These two surveys were carried out using similar methodologies but involved different populations, resulting in independent samples for each year. The NHS is a household health survey developed with the scope of health surveillance and assistance in partnership with the Ministry of Health, the Oswaldo Cruz Foundation, and the Brazilian Institute of Geography and Statistics (IBGE) [11]. The databases and variables dictionaries for 2013 and 2019 were obtained from the IBGE website in June 2022. The database version contained updates and corrections made to the 2013 (updated on 25 August 2020) and 2019 (updated on 24 May 2022) surveys. These included corrections for sample weight based on the Population Projection of the Federation Units by sex and age for 2010–2060.

In each year of the NHS, the sample was selected from residents residing in permanent private households across Brazilian urban and rural areas, encompassing five geographic macro-regions, federative units, capitals, and metropolitan regions [12]. The NHS sample is a subsample of the IBGE Master Sample, used as unities of many areas selected to be used in several national surveys. To determine the sample size required for estimating parameters of interest across various levels of geographic disaggregation in the NHS, several factors were taken into account: the estimated proportions and the desired level of precision within 95% confidence intervals (95% CI), the design effect (Deff) due to the multi-stage cluster sampling method used, the number of households selected per primary sampling unit (PSU), and the proportion of households containing individuals within the target age group [13]. The sample selection occurred in three stages using a simple random draw: census sectors or sets of sectors (PSU), permanent private households (Second Sampling Stage), and adults residing in these households (Third Sampling Stage). For each PSU, 10 to 14 households were randomly selected, depending on the domain size, to reach the minimum sample required. Within each household, one resident was chosen randomly with equal probability among eligible participants. Sampling weights were defined for the PSU, households, and all their residents. Further details on the sampling procedure and weighting factors can be found in previous publications [12,13,14]. The ages of interest were over 18 years in the 2013 NHS [14] and over 15 years in the 2019 NHS [11]. For this study, data for adults under 18 years from the NHS 2019 [14] were excluded to allow comparison between surveys [10].

Data were collected through interviews with randomly selected participants using a structured questionnaire comprising three sections: household information, all household residents, and individual characteristics of the selected respondent. The questionnaire was administered by interviewers who were trained on the survey methodology and the materials to be used during the interview. This study specifically analyzed individual variables of the selected resident obtained from modules C—General characteristics of residents, D—Education of residents aged 5 or over, E—Job of household residents, F—Household income, and U—Oral health (Appendix A). The modules and questions used in the PNS were the same in both years, allowing data analysis in the period.

The outcome was the SROH, which was classified as either positive or negative. Positive SROH corresponded to very good and good responses to the question “In general, how would you rate your oral health (teeth and gums)?”. Negative SROH included regular, bad, and very bad responses. Previous studies have also categorized SROH into positive and negative, with regular responses in the negative category [2,15]. Vieira et al. [15], in their analysis of factors associated with SROH, found that the variation in proportions for variables among individuals with fair oral health self-perception closely resembled those observed in individuals with a negative oral health self-perception (poor/very poor).

Socioeconomic indicators included education and income. Residents’ responses to the following questions were considered to assess education: frequent school (Yes/No), which course they frequented, previously frequented school, and the highest course they frequented. Based on the Brazilian school system, schooling was converted into years of study according to the following categories used in previous studies [4,7]: 0 to 4 (never frequented school, nursery, preschool, youth and adults literacy, youth and adults education [EJA], or supplementary elementary education); 5 to 8 (regular course of elementary education); 9 to 11 (regular course of high school or EJA or supplementary high school); and 12 or more years of study (higher education—undergraduate, higher-level specialization, masters, or PhD). Per capita income was calculated by dividing the total household income—comprising gross income from the main job, income in goods and products, earnings from secondary jobs (both in money and goods), retirement benefits, alimony, rent, and interest savings account—by the number of residents in the household. Per capita income was converted into minimum wages (MWs) (2013: BRL 678.00—USD 332.00 and 2019: BRL 998.00—USD 261.00) and categorized into 0–1 MWs, 1.1–2 MWs, 2.1–3 MWs, 3.1 or more MWs, according to previous studies.

The covariates were sex (male; female) and age, with age groups categorized as 18–24, 25–39, 40–59, and over 60 years old [1].

The descriptive analysis was performed to describe the total sample according to income, education, sex, and age group. Positive SROH prevalence was estimated for the total sample and according to income, education, sex, and age group. The prevalence of SROH was also estimated for each Brazilian region, considering socioeconomic groups, and results are shown in bar graphs for 2013 and 2019. Additionally, we employed a logistic regression model to investigate the association between income and education with SROH adjusted for region, sex, and age group. In this model, we also examined the interaction between income and education. Calculating the marginal estimates, we obtained the adjusted prevalence of positive SROH in Brazil for each survey (2013 and 2019) according to income and education levels. The theory of the social determinants of oral health was employed to guide the analysis of inequalities shaped by socioeconomic indicators [5,16].

The magnitude of inequalities in SROH among education and income groups was analyzed using the Slope Index of Inequality (SII) and Relative Index of Inequality (RII). The SII and RII are summary measures recommended for comparisons across populations [17]. These indices are regression-based and consider the entire socioeconomic distribution rather than only comparing the two most extreme groups. The midpoint was calculated by ordering the social groups from lowest to highest. The population of each social group category covers a range in the cumulative distribution of the population and is given a score based on the midpoint of its range in the cumulative distribution of the population. The SII can then incorporate changes in the distribution of social groups over time that affected the population health burden of health disparities between 2013 and 2019 [17]. The SII can be interpreted as the absolute difference in health outcomes between the top and bottom socioeconomic groups as defined by income and education categories. The RII was interpreted as the ratio of health outcomes between groups. If there is no inequality, the SII assumes a value of zero. Positive SII values indicate a higher prevalence of positive SROH in the group with greater social advantage. Larger values indicate greater disparity magnitudes. If there is no inequality, the RII assumes a value of 1.0. The further the value of RII is from 1.0, the higher the level of inequality. RII assumes only positive values, with values larger than one indicating a concentration of positive SROH among the advantaged and values smaller than one indicating a concentration of the outcome among the disadvantaged. Socioeconomic differences between 2013 and 2019 were tested using a two-way interaction term ridit score by survey [18,19]. A positive and significant coefficient for the interaction term indicates an increase in the SII (or RII) between the groups.

We used generalized linear models (log-binomial regression) with an identity link function to calculate the SII (rate differences) and a logarithmic link function to calculate the RII (rate ratios). Both indices were estimated with 95% confidence intervals and were adjusted for sex and age. Statistical significance was set at *p* < 0.05. All analyses were performed using the Stata statistical package, version 18.0 (StataCorp LP, College Station, TX, USA), accounting for complex survey design and sampling weights employed using the “svy” command.

The 2009 (CAAE: 10853812700000008) and 2013 (CAAE: 11713319700000008) NHS projects were approved by the National Commission of Ethics in Research (CONEP). All participants signed the informed consent.

## 3. Results

In 2013, 64,308 adults participated in the NHS, with 93.62% responding to the oral health module and sociodemographic variables. Of 60,202 individuals, 11 (0.02%) did not respond to the income variable. In 2019, 88,531 adults responded to the oral health module and sociodemographic variables out of 90,846 respondents. Of these, 22 (0.02%) did not respond to income variables. In 2013, 52.90% were women, 34.24% were between 40 and 59 years old, 34.30% had 9–11 years of study, and 49.74% had per capita income between 0 and 1 MWs. In 2019, 53.16% were women, 35.30% were between 40 and 59 years old, 35.91% had between 9 and 11 years of study, and 51.24% had per capita income up to 1 MWs (Table 1). Women and younger adults (18–24 years old) had a higher frequency of positive SROH in both surveys, as well as individuals more educated (≥12 years) and with higher incomes (≥3.1 MWs) (Table 1).

In Brazil, the prevalence of positive SROH was 67.50% and 69.68% in 2013 and 2019, respectively. A higher prevalence of positive SROH was observed among individuals with higher education (2013: 82.55; 2019: 82.15) and income (2013: 84.50; 2019: 83.9) levels compared to those with lower socioeconomic levels (education—2013: 57.11; 2019: 61.99; income—2013: 59.98; 2019: 62.93). There was a higher prevalence of positive SROH in individuals with lower education and income levels in 2019 than in 2013. The prevalence remained almost unchanged in the higher education and income group in 2019 compared to 2013 (Table 1).

Figure 1 and Figure 2 illustrate the prevalence of positive SROH according to education and income across each Brazilian region in 2013 and 2019. In 2013 and 2019, the north and northeast regions exhibited the lowest prevalences of positive SROH among individuals with the lowest income and education attainment. By 2019, the prevalence of positive SROH among those with the lowest income and education in these regions had increased compared to 2013 and approached levels observed in other Brazilian regions. In the north, northeast, and southeast regions, the prevalence of positive SROH in 2019 was higher among groups with the lowest education (0–4 years of education) compared to 2013. In the north region, differences were noted between groups with 5–8 years and 9–11 years of education (Figure 1). Similar findings were observed concerning income in the north and northeast regions. There was a higher prevalence of positive SROH among those with low income in 2019 compared to the same income group in 2013. No changes in the prevalence of positive SROH were observed among groups with higher income and education levels in 2019 compared to 2013 across any Brazilian region (Figure 2).

The logistic regression models are presented in the Appendix A. A significant interaction between income and education was observed in 2019. Marginal estimates from the logistic regression model indicate that individuals with higher social advantage—greater education and income—showed the highest estimated prevalence of positive SROH in both 2013 and 2019 (Figure 3). Adults with ≥12 years of education exhibited a higher prevalence of positive SROH, even when their income was lower, compared to those with similar income but fewer years of education across both years. Conversely, individuals with lower income demonstrated a lower prevalence of positive SROH regardless of education level. In 2013, the association between income and positive SROH appeared consistent across all education categories. However, by 2019, adults with 5–8 years of education and an income of 2.1–3 MWs exhibited a higher prevalence of positive SROH than those with incomes of 0–1 MWs or 1.1–2 MWs. This pattern was not significantly different among those with 0–4 and 9–11 years of education. For individuals with 12 or more years of education, the prevalence of positive SROH was higher than that observed in individuals with lower incomes (Figure 3). The adjusted models are detailed in the Appendix A.

Table 2 presents the SII and RII values for Brazil and each region, along with interaction terms and corresponding *p*-values. The SII values were positive, and RII values exceeded 1.0 for both socioeconomic indicators across all regions in 2013 and 2019. These results reaffirm a higher prevalence of positive SROH among individuals with greater social advantage (higher income and higher education). A decrease in the magnitude of absolute and relative education-based inequality in SROH between 2013 and 2019 was observed for the total Brazilian sample. Regional analysis showed a reduction in relative education-based disparities in the north and northeast regions. However, for income disparities, a decrease in the magnitude of the Relative Inequality Index (RII) was observed only in the north region (Table 2).

## 4. Discussion

The results indicated disparities in SROH among social groups in all Brazilian regions, with a higher prevalence of positive SROH among those with higher income and education, demonstrating the persistence of the social gradient of SROH in the country in the last decade. Despite the persistence of inequalities, a reduction in the magnitude of relative education-based inequalities was observed in Brazil, and relative income-based inequalities were observed in the north region over the years.

The findings demonstrate the importance of education and income as indicators that capture distinct facets of socioeconomic advantages or disadvantages on health outcomes. Individuals with low levels of both socioeconomic indicators exhibited the lowest prevalence of positive SROH. This vulnerable group faces numerous barriers and lacks the resources necessary for optimal health. Education and income are indicators of socioeconomic status. They are reportedly associated with the risk of illness and mortality when individuals with lower education and income levels present a greater chance of getting sick [2,3,4,5,6,10,16]. When analyzed in concert, income and education serve as complementary metrics for defining the multidimensional construct of socioeconomic status. Investigators have suggested that the observed health effects of education and income may serve as proxies for disparities in employment opportunities, housing conditions, access to nutritious foods, and health insurance coverage. Individuals with multiple adverse social determinants, including limited education and low income, are potentially at a heightened risk of adverse health outcomes, such as those encompassed by the SROH [2,3,16,20]. When exploring the incidence of coronary disease in the United States, Lewis et al. [20] found that the combined presence of low income and lower education is associated with a greater risk of coronary disease compared with low income or low education separately.

The findings of this study suggest that equitable Brazilian oral health policies can reduce SROH inequality over time. The magnitude of the relative disparity in SROH decreased according to education levels in Brazil, mainly because of the decrease in the relative magnitude of disparity in education within the northeast and north regions. These areas experienced increased positive SROH prevalence among less-educated groups from 2013 to 2019, with minimal changes in more privileged groups. A reduction in income inequality was observed only in the north region. This shift may reflect enhanced access to health services within the SUS for socially disadvantaged groups owing to equitable and universal public policies, ultimately improving the oral health of Brazilian adults [8]. In addition, these findings should be interpreted in light of the social transformations that have occurred since 2003, mainly because of the implementation of redistributive policies, which have had some positive effects on socioeconomic, health, and oral health indicators, as broad social benefit coverage significantly reduces oral health inequalities [8]. These policies, such as the Bolsa Família program and the Continuous Payment Benefit program, have positively affected poverty and income inequality, reducing the number of impoverished people in regions such as the northeast and north, the poorest regions of the country [21]. The implementation of Brazil’s NOHP promoted the increased provision and coverage of public dental services from 2003 to 2006, which continued to some degree in the subsequent years. The percentage of people who never used dental services decreased over time, indicating improved coverage of dental services in the country [8]. Education interventions and increased access to information may raise awareness among individuals with lower education levels about the importance of oral health, as education is a potent tool in breaking the cycle of poverty and promoting health equity [21]. Individuals with higher incomes and education levels tend to seek and use health services better. The Atlas for Human Development of 2013 showed an increase in the mean years of education in the Brazilian adolescent and adult populations, positively affecting socioeconomic, health, and oral health outcomes [21]. Changes in the labor market in Brazil demonstrated a reversal in the unemployment and informal employment trends, which, although not homogenous, improved, particularly in the northeast region of Brazil [22].

This study aimed to illustrate the distribution of inequalities in oral health among Brazilians and evaluate changes over time as potential indicators of the influence of social policies, specifically redistributive and NOHP, on oral health inequalities (SROH). The assessment of the changes in oral health status over time involves the description of change; identification of differences among social groups in terms of nature, direction, and magnitude of change; identification of predictors of changes, such as education and income; and determination of an explanation of the change. Using absolute (SII) and relative (RII) measures is essential for assessing the magnitude of inequalities since how inequalities are measured impacts the understanding of whether inequalities are improving or worsening with time. In our study, there was a significant decrease in the magnitude of relative education-based inequalities in positive SROH in Brazil, mainly due to the improvement in SROH among the less-educated individuals from the north and northeast regions. Income-based inequalities showed a significant decrease only in the north region, perhaps because subjective measures such as the SROH are more sensitive to knowledge than income measures. This finding aligns with Farmer et al. [23], who compared the contribution of education and income to two oral health outcomes: chewing difficulties and SROH. They found that education played a larger role than income in explaining inequalities in SROH because it is a subjective measure.

Oral health inequalities are a global challenge [9,16]. Despite these improvements, persistent income and education-based inequalities have been observed in Brazil. Persistent disparities align with findings indicating that individuals with higher socioeconomic levels generally experience better health outcomes, including a higher prevalence of functional dentition and lower levels of dental caries, periodontal disease, and tooth loss than their counterparts [18,24,25,26,27,28,29]. Similarly to the findings of this study, when comparing edentulism in adults and elders and the effect of dental services utilization in Brazil, Ferreira et al. [28] found that complete tooth loss was concentrated among disadvantaged subgroups in terms of income and education. The use of dental services mitigated the harmful effects of social disadvantage among adults and reduced the extremes of the education hierarchy [28]. In a comparison of education-related oral health inequalities in older adults in Japan and Singapore, Kiuchi et al. [29] also found a significant association between being educated and lack of functional dentition in both countries. Singapore exhibited higher education-related relative inequalities (RII) and absolute inequalities (SII) compared to Japan [29]. These differences between the top and bottom of society result from health knowledge, literacy, healthier behaviors, improved healthcare access, and the influence of prestige and labor market opportunities [2,3,6,16,24,25]. Karam et al. [4] showed that limited education and financial constraints could hinder access to oral health counseling, healthy diets, and dental service information. Investing in education shows promise for reducing Brazilian health disparities because enhancing oral health literacy can boost oral health knowledge [26]. Then, addressing these inequalities requires intersectoral policies, improved access to health information, and focusing on underprivileged groups through education and social programs to avoid the “inverse equity” hypothesis [26], where public health efforts benefit the affluent more. These actions must be supported by a global health network that develops a cost-effective oral health system, incorporates oral health into the broader healthcare agenda, and guides relevant policy development [9].

The strengths of this study are that the data were obtained from nationally representative health surveys and included education and income levels, which are the most common proxies of social position for measuring absolute and relative inequalities. Given the sample calculation, the findings are representative of the population of Brazil as a whole and of each Brazilian region. To the best of our knowledge, this is the first study to evaluate changes in the magnitude of socioeconomic inequalities in SROH among Brazilian people using a two-way interaction term ridit score to analyze modifications in SII and RII over time. The design of the NHS excludes the homeless population and residents of long-stay institutions. In addition, the interviews were carried out with only one resident. While it is acknowledged that conducting a gender analysis is important because men and women may have different risk factors, access to care, biological influences, and social determinants that impact their oral health outcomes, this study opted to adjust the estimates only by sex. Future research could stratify by sex to ensure a comprehensive understanding of oral health disparities and promote more equitable healthcare for all. Some measurement bias may have occurred because socioeconomic indicators were self-reported. The subjectivity embedded in the evaluation of SROH is influenced by circumstances in a person’s life, day, and week and is a result of the contextual and psychosocial conditions experienced by the individual, involving values and feelings that are not expressed. This characteristic provides a comprehensive assessment of how people perceive their oral health and its effects of oral health on the functional, social, and psychosocial aspects of daily life. Subjectivity qualifies the relevance of the findings to health policies and decision-making. Despite our limitations, our results represent the inequalities in SROH regarding income and education in Brazil. Research and policies that focus on a more equitable distribution of power, prestige, opportunities, and resources in income and education could improve health conditions and alleviate the negative perceptions of oral health among marginalized individuals.

## 5. Conclusions

A higher prevalence of positive SROH was observed among individuals who accumulated social advantages characterized by high income and education levels. The prevalence of positive SROH increased in groups with lower education levels from 2013 to 2019 in the north and northeast regions, resulting in a decrease in the magnitude of education-based inequality in SROH. Furthermore, income-based inequality has been reduced in the north region of Brazil. The findings of this study suggest that equitable Brazilian oral health policies can reduce SROH inequality over time.

## Figures and Tables

**Figure 1 ijerph-21-01198-f001:**
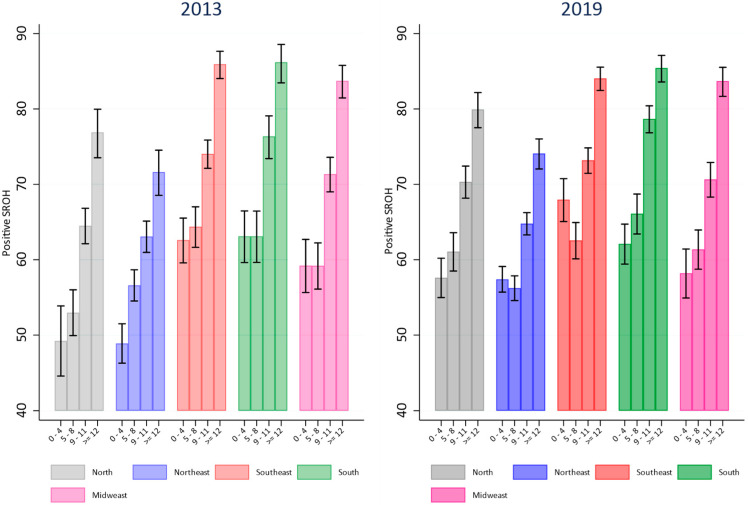
Comparison of prevalence of positive SROH according to education levels in Brazilian regions in 2013 and 2019.

**Figure 2 ijerph-21-01198-f002:**
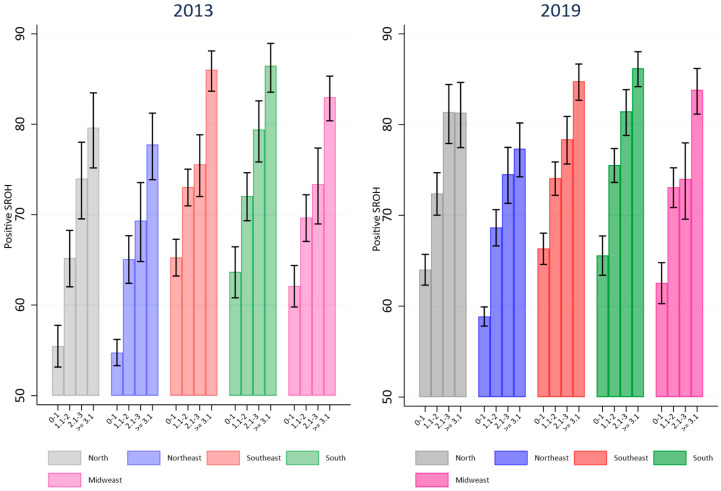
Comparison of prevalence of positive SROH according to income levels in Brazilian regions in 2013 and 2019.

**Figure 3 ijerph-21-01198-f003:**
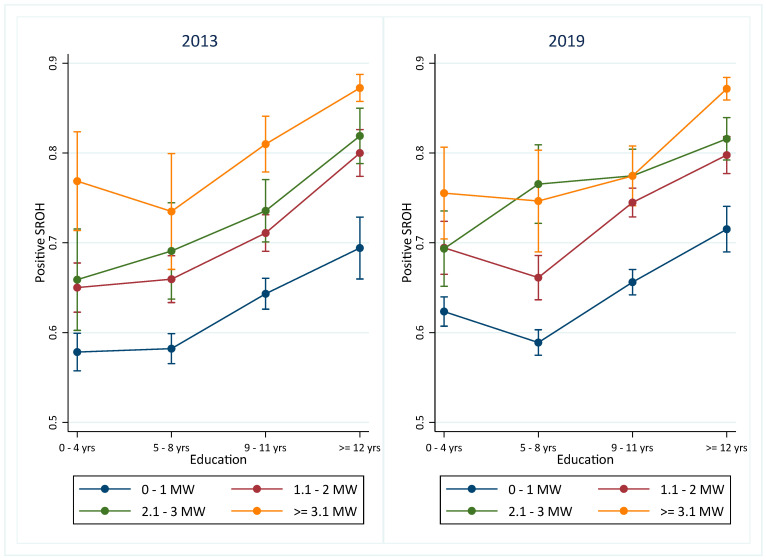
Marginal estimates of positive SROH from adjusted logistic regression model according to income and education in 2013 and 2019. Note: The x-axis in the graph shows the education categories, and the different colors of the lines represent the income categories.

**Table 1 ijerph-21-01198-t001:** Distribution of the total sample of adults and those with positive SROH by sex, age group, education, and income in 2013 and 2019 in Brazil.

	2013 (*n* = 60,202)		2019 (*n* = 88,531)	
	Total Sample	Positive SROH	Total Sample	Positive SROH
	*n* (%, 95% CI)	*n* (%, 95% CI)	*n* (%, 95% CI)	*n* (%, 95% CI)
*Sex*				
Male	25,920 (47.10, 46.43; 47.87)	16,570 (65.95, 64.90; 66.99)	41,662 (46.84, 46.24; 47.44)	27,462 (68.29, 67.42; 69.14)
Female	34,282 (52.90, 52.13; 53.66)	23,002 (68.88, 67.94; 69.80)	46,869 (53.16, 52.56; 53.76)	32,423 (70.91, 70.13; 71.68)
*Age group (years old)*				
18–24	7823 (15.93, 15.37; 16.50)	5689 (74.43, 72.70; 76.09)	8145 (13.87, 13.36; 14.39)	5998 (75.63, 73.93; 77.25)
25–39	20,767 (31.77, 31.09; 32.47)	14,419 (71.17, 70.06; 72.25)	25,339 (29.23, 28.64; 29.81	18,149 (72.84, 71.88; 73.77)
40–59	20,435 (34.24, 33.59; 34.91)	12,730 (63.57, 62.34; 64.78)	32,259 (35.30, 34.71; 35.89)	21,015 (66.68, 65.63; 67.70)
≥60	11,177 (18.06, 17.48; 18.65)	6734 (62.38, 60.72; 64.01)	22,728 (21.61, 21.08; 22.16)	14,723 (66.51, 65.43; 67.56)
*Education (years of study)*				
0–4	13,139 (21.43, 20.77; 22.11)	7192 (57.11, 55.51; 58.71)	19,499 (18.11, 17.66; 18.57)	11,715 (61.99, 60.71; 63.26)
5–8	15,239 (25.09, 24.39; 25.81)	8970 (60.61, 59.20; 62.01)	21,736 (23.34, 22.78; 23.91	13,167 (60.89, 59.71; 62.06)
9–11	20,026 (34.30, 22.56; 35.04)	13,840 (70.61, 69.49; 71.70)	28,552 (35.91, 35.30; 36.53)	19,817 (71.41, 70.49; 72.32)
≥12	11,798 (19.18, 18.38; 20.01)	9570 (82.55, 81.30; 83.74)	18,744 (22.64, 21.90; 23.39)	15,186 (82.15, 81.18; 83.08)
*Income (minimum wage)*				
0–1	31,760 (49.74, 48.80; 50.68)	18,668 (59.98, 58.97; 60.97)	48,303 (51.24, 50.43; 52.05)	29,778 (62.93, 62.14; 63.71)
1.1–2	15,493 (19.18, 18.38; 20.01)	10,647 (70.74, 69.46; 71.98)	22,153 (28.16, 27.56; 28.78)	15,691 (73.33, 72.23; 74.39)
2.1–3	5335 (9.34, 8.90; 9.80)	3974 (75.45, 73.30; 77.47)	7515 (9.07, 8.73; 9.43)	5742 (78.37, 76.74; 79.91)
≥3.1	7603 (12.12, 11.42; 12.85)	6272 (84.50, 83.00; 85.90)	10,538 (11.52, 10.98; 12.13)	8655 (83.90, 82.59; 85.12)

**Table 2 ijerph-21-01198-t002:** Education and income-based inequalities in positive SROH in Brazil and Brazilian regions (NHSs 2013 and 2019).

	Education-Based Inequality			
	SII (95% CI) *	SII (95% CI) *	Interaction term *	*p*
	2013	2019		
BRAZIL	0.31 (0.29; 0.34)	0.28 (0.26; 0.30)	−0.04 (−0.07; −0.01)	0.005
North	0.33 (0.26; 0.40)	0.26 (0.19; 0.31)	−0.07 (−0.15; 0.11)	0.092
Northeast	0.24 (0.19; 0.30)	0.22 (0.18; 0.25)	−0.05 (−0.11; 0.00)	0.063
Southeast	0.30 (0.25; 0.34)	0.25 (0.22; 0.29)	−0.05 (−0.10; 0.01)	0.077
South	0.31 (0.25; 0.36)	0.29 (0.25; 0.33)	−0.02 (−0.08; 0.04)	0.488
Midwest	0.32 (0.27; 0.38)	0.31 (0.27; 0.36)	−0.01(−0.07; 0.06)	0.896
	RII (95% CI) *	RII (95%CI) *	Interaction term *	*p*
	2013	2019		
BRAZIL	1.58 (1.52; 1.65)	1.48 (1.44; 1.52)	0.93 (0.89; 0.97)	0.001
North	1.70 (1.51; 1.93)	1.45 (1.35; 1.56)	0.85 (0.74; 0.97)	0.018
Northeast	1.50 (1.38; 1.64)	1.41 (1.33; 1.48)	0.89 (0.81; 0.98)	0.016
Southeast	1.49 (1.40; 1.59)	1.40 (1.33; 1.48)	0.93 (0.86; 1.00)	0.065
South	1.51 (1.39; 1.64)	1.50 (1.42; 1.58)	0.99 (0.91; 1.08)	0.831
Midwest	1.58 (1.46; 1.71)	1.55 (1.45; 1.65)	0.99 (0.90; 1.09)	0.844
	Income-Based Inequality			
	SII (95% CI) *	SII (95% CI) *	Interaction term *	*p*
	2013	2019		
BRAZIL	0.35 (0.32; 0.37)	0.32 (0.30; 0.34)	−0.03 (−0.06; 0.00)	0.093
North	0.34 (0.28; 0.40)	0.29 (0.24; 0.34)	−0.05 (−0.12; 0.03)	0.207
Northeast	0.32 (0.27; 0.36)	0.29 (0.26; 0.32)	−0.02 (−0.08; 0.03)	0.445
Southeast	0.28 (0.24; 0.33)	0.28 (0.24; 0.31)	−0.01 (−0.06; 0.05)	0.827
South	0.33 (0.27; 0.38)	0.31 (0.27; 0.35)	0.01 (−0.06; 0.08)	0.745
Midwest	0.29 (0.24; 0.34)	0.31 (0.26; 0.36)	0.02 (−0.05; 0.09)	0.600
	RII (95% CI) *	RII (95% CI) *	Interaction term *	*p*
	2013	2019		
BRAZIL	1.64 (1.58; 1.70)	1.56 (1.52; 1.60)	0.96 (0.92; 1.00)	0.064
North	1.71 (1.55; 1.89)	1.51 (1.42; 1.62)	0.87 (0.77; 0.97)	0.015
Northeast	1.66 (1.54; 1.78)	1.55 (1.47; 1.62)	0.95 (0.87; 1.03)	0.229
Southeast	1.46 (1.37; 1.56)	1.44 (1.37; 1.52)	0.99 (0.92; 1.07)	0.832
South	1.54 (1.43; 1.67)	1.53 (1.44; 1.63)	0.99 (0.90; 1.10)	0.892
Midwest	1.50 (1.39; 1.61)	1.55 (1.44; 1.66)	1.04 (0.93; 1.15)	0.505

* The analyses were adjusted for sex and age, accounting for complex survey design and sampling weights.

## Data Availability

The database that supports the findings of this study is available at https://www.pns.icict.fiocruz.br/bases-de-dados/ accessed on 6 June 2024.

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
