# Peer review of "Decrease in Socioeconomic Disparities in Self-Rated Oral Health among Brazilian Adults between 2013 and 2019: Results from the National Health Survey"

_ijerph, 2024, doi:10.3390/ijerph21091198_

Round 1
Reviewer 1 Report
Comments and Suggestions for Authors
The topic should be "Decreasing socioeconomic disparities in self-rated oral health: National Health Survey among Brazilian adults between 2013 and 2019 using Meta-Analysis". Please ensure the abstract includes the objective and method (data collection and analysis, such as sampling technique and statistical analysis). The last paragraph of the introduction is necessary to add the research gap in Brazil. Definitions of terms, theories, and variables in this study are necessary to be added. The researchers mentioned interviews but no results for this method. The table and figures in the results show the evidence of meta-analysis; if yes, please add the methodology of meta-analysis. It is necessary to discuss the theories and variables adopted in this study. And how findings support or argue with previous studies, please identify. The conclusion could be more comprehensive to be useful for strategic planners and further studies. References could be more up-to-date in 2023 and 2024.
Comments on the Quality of English LanguageMinor changes by English professionals are required.
Author Response
Reviewer #1
Reviewer #1 – point 1: The topic should be "Decreasing socioeconomic disparities in self-rated oral health: National Health Survey among Brazilian adults between 2013 and 2019 using Meta-Analysis".
Author response: Thank you for the title suggestion. However, this study did not use meta-analysis. The presented graphs show the values of the coefficients from the logistic regression model, as an alternative to using a table (coefplot). In this regard, we believe the suggested title does not accurately reflect what is presented in the manuscript.
Reviewer #1 – point 2: Please ensure the abstract includes the objective and method (data collection and analysis, such as sampling technique and statistical analysis).
Author response: In the abstract, we present the study objective "This cross-sectional study assessed the magnitude of inequalities in self-rated oral health (SROH) among different socioeconomic groups in Brazil." The data were collected through interviews with the participants, as stated in the abstract. We highlighted the method of analysis used to measure the magnitude of oral health inequalities in the abstract because it is the central focus of the study. Regarding the sample, it was mentioned that data from the National Health Survey in 2013 and 2019 were analyzed, with their respective sample sizes. More details about the sample are provided in the Materials and Methods section of the manuscript.
Abstract: This cross-sectional study assessed the magnitude of inequalities in self-rated oral health (SROH) among different socioeconomic groups in Brazil. Secondary data from interviews with a sample of adults (> 18 years) from the National Health Survey 2013 (n=64,308) and 2019 (n=88,531) were analyzed. Positive SROH was considered when participants selected the good or very good options. Socioeconomic indicators were monthly household income and years of education. The magnitude of inequalities among socioeconomic groups was estimated using the Slope (SII) and Relative Index of Inequality (RII). Interaction term assessed changes in SII/RII over time. Estimates were adjusted for sex and age. The prevalence of SROH was 67.5% in 2013 and 69.7% in 2019. Individuals with lower socioeconomic indicators had a lower prevalence of positive SROH. Significant reductions in the magnitude of the education-based RII between 2013 (1.58) and 2019 (1.48) in Brazil, as well as in North (1.70; 1.45) and Northeast (1.50; 1.41) regions and reduction in the income-based RII in North (1.71; 1.51) were observed. Socioeconomic inequalities in SROH persist across different Brazilian regions, although there was a reduction in disparities among educational groups in 2019 compared with 2013. The findings of this study suggest that equitable Brazilian oral health policies may have contributed to reducing SROH inequality over time.
Reviewer #1 – point 3: The last paragraph of the introduction is necessary to add the research gap in Brazil.
Author response: Thank you for your comment. We have added a phrase to the last paragraph of the introduction to highlight the research gap.
Despite the significant progress made, there is still a lack of comprehensive studies that evaluate the long-term effect of the NOHP in reducing oral health inequalities among different socioeconomic and regional groups in Brazil. This study addresses this gap by providing a detailed analysis of how disparities in oral health have evolved. Then, this study aimed to monitor disparities in oral health about a decade after the implementation of the NOHP, with a second evaluation conducted six years later. Recognizing the vast geographical, economic, and cultural diversity in the country, the objective was to investigate the magnitude of disparities in positive SROH among socioeconomic groups in Brazil between 2013 and 2019 and across Brazilian regions. Our hypothesis is that socioeconomic disparities decreased between 2013 and 2019.
Reviewer #1 – point 4: Definitions of terms, theories, and variables in this study are necessary to be added.
Author response: The analyzed variables were described in 4th and 5th paragraphs of the Materials and Methods section. We adopted the theory of the social determinants of oral health to guide the analysis of inequalities as shaped by socioeconomic indicators. This information was added in the 7th paragraph of the Material and Methods section.
The outcome was the SROH, which was classified as either positive or negative. Positive SROH corresponded to very good and good responses to the question “In general, how would you rate your oral health (teeth and gums)?”. Negative SROH included regular, bad, and very bad responses. Previous studies have also categorized SROH into positive and negative, with regular responses in the negative category [2, 14]. Vieira et al. [14], in their analysis of factors associated with SROH, found that the variation in proportions for variables among individuals with fair oral health self-perception closely resembled those observed in individuals with a negative oral health self-perception (poor/very poor).
Socioeconomic indicators included education and income. Residents' responses to the following questions were considered to assess education: frequent school (Yes/No), which course they frequented, previously frequented school, and what the highest course they frequented. Based on the Brazilian school system, schooling was converted into years of study according to the following categories used in previous studies [4, 7]: 0 – 4 (never frequented school, nursery, preschool, youth and adults literacy, youth and adults education [EJA] or supplementary elementary education); 5 to 8 (regular course of elementary education); 9 to 11 (regular course of high school or EJA or supplementary high school); and 12 or more years of study (higher education - undergraduate, higher-level specialization, master or PhD). Per capita income was calculated by dividing the total household income - comprising gross income from the main job, income in goods and products, earnings from secondary jobs (both in money and goods), retirement benefits, alimony, rent, and interest savings account - by the number of residents in the household. Per capita income was converted into minimum wages (MW) (2013: R$678.00 - US$332.00 and 2019: R$998.00 - US$261.00) and categorized into 0 – 1 MW; 1.1 – 2 MW; 2.1 – 3 MW; 3.1 or more MW, according to previous studies.
The covariates were sex (male; female) and age, with age groups categorized as 18 – 24, 25-39, 40 – 59, and over 60 years old [1].
The descriptive analysis was performed to describe the total sample according to income, education, sex, and age group. Positive SROH prevalence was estimated for the total sample and according to income, education, sex, and age group. The prevalence of SROH was also estimated for each Brazilian region, considering socioeconomic groups, and results were shown in bar graphs for 2013 and 2019. Additionally, we employed a logistic regression model to investigate the association between income and education with SROH adjusted for region, sex, and age group. In this model, we also examined the interaction between income and education. Calculating the marginal estimates, we obtained the adjusted prevalence of positive SROH in Brazil for each survey (2013 and 2019) according to income and education levels. The theory of the social determinants of oral health was employed to guide the analysis of inequalities shaped by socioeconomic indicators [5, 16].
Reviewer #1 – point 5: The researchers mentioned interviews but no results for this method.
Author response: We would like to clarify that all variables analyzed in this study were collected through interviews with participants in their residences, using a structured questionnaire, as mentioned in the Materials and Methods section of the manuscript. Thus, the first paragraph of the Results section presents the descriptive results of these variables to outline the profile of the investigated sample.
Reviewer #1 – point 6: The table and figures in the results show the evidence of meta-analysis; if yes, please add the methodology of meta-analysis.
Author response: Thank you for your comments. As mentioned in the first point, this study did not use meta-analysis. The findings are results from logistic regression model. Thus, we believe it is not necessary to include information of meta-analysis on the methodology of study.
Reviewer #1 – point 7: It is necessary to discuss the theories and variables adopted in this study. And how findings support or argue with previous studies, please identify.
Author response: Thank you for your suggestion. On the discussion section, we presented previous studies that demonstrated the persistence of inequalities in Oral Health and the possible theories to reduce the disparities across different social groups, as showed bellow.
Oral health inequalities are a global challenge [9, 16]. Despite these improvements, persistent income and education-based inequalities have been observed in Brazil. Persistent disparities align with findings indicating that individuals with higher socioeconomic levels generally experience better health outcomes, including a higher prevalence of functional dentition and lower levels of dental caries, periodontal disease, and tooth loss than their counterparts [18,24-29]. Similarly to the findings of this study, when comparing edentulism in adults and elders and the effect of dental services utilization in Brazil, Ferreira et al. [28] found that complete tooth loss was concentrated among disadvantaged subgroups in terms of income and education. The use of dental services mitigated the harmful effects of social disadvantage among adults and reduced the extremes of education hierarchy [28]. In a comparison of education-related oral health inequalities in older adults in Japan and Singapure, Kiuchi et al. [29] also found a significant association between being edentate and lack of functional dentition in both countries. Singapure exhibited higher education-related relative inequalities (RII) and absolute inequalities (SII) compared to Japan [29]. These differences between the top and bottom of society result from health knowledge, literacy, healthier behaviors, improved healthcare access, and the influence of prestige and labor market opportunities [2,3,6,16, 24,25]. Karam et al. [4] showed that limited education and financial constraints could hinder access to oral health counseling, healthy diets, and dental service information. Investing in education shows promise for reducing Brazilian health disparities because enhancing oral health literacy can boost oral health knowledge [26]. Then, addressing these inequalities requires intersectoral policies, improved access to health information, and focusing on underprivileged groups through education and social programs to avoid the “inverse equity” hypothesis [26], where public health efforts benefit the affluent more. These actions must be supported by a global health network that develops a cost-effective oral health system, incorporates oral health into the broader healthcare agenda, and guides relevant policy development [9].
Reviewer #1– point 8: The conclusion could be more comprehensive to be useful for strategic planners and further studies.
Author response: Thank you for your suggestion. We have included some reflections on the focus of new research and policies to address inequalities, which may be useful for researchers and decision-makers.
(…) This characteristic provides a comprehensive assessment of how people perceive their oral health and its effects of oral health on the functional, social, and psychosocial aspects of daily life. Subjectivity qualifies the relevance of the findings to health policies and decision-making. Despite our limitations, our results represent the inequalities in SROH regarding income and education in Brazil. Research and policies that focus on a more equitable distribution of power, prestige, opportunities, and resources in income and education could improve health conditions and alleviate the negative perceptions of oral health among marginalized individuals.
Reviewer #1 – point 9: References could be more up-to-date in 2023 and 2024.
Author response: Thank you for your suggestion. Considering the update of our refereces, we included in our manuscript findings of studies that analysed inequalities of oral health outcomes (edentulism, functional dentition) in Brazil and around the world, and the effect of the socioeconomic indicators and use of services in the inequalities.

Reviewer 2 Report
Comments and Suggestions for Authors
Thank you for the opportunity to review this manuscript. The authors present a cross-sectional analysis of data from the 2013 and 2019 National Health Survey, with large and representative samples, which explores socioeconomic inequalities in self-rated oral health. The introduction provides a comprehensive overview of existing literature and highlights the ways in which socioeconomic factors contribute to oral health, particularly in context of the National Oral Health Policy that was introduced in Brazil in 2004 and sought to reduce inequalities in oral health. The authors find that individuals with lower socioeconomic indicators had a lower prevalence of positive self-rated oral health in both 2013 and 2019, highlighting the persistence of the social gradient for self-rated oral health. Further, they show that a reduction in the magnitude or relative education-based and income-based inequalities have been observed in some regions in Brazil and this is associated with positive self-rated oral health. The findings have important implications for equitable policies which seek to reduce self-rated oral health inequalities. This is a very well-written manuscript, with appropriate statistical analyses, and novel findings. I have only minor suggestions for revisions:
· 1. In the methods section, it could be clearer whether the National Health Survey is a repeated cross-sectional study, i.e., did the study ask the same questions in 2013 and 2019? And were different individuals recruited for each study?
· 2. The statistical analyses are described in great detail and the use of the Slope Index of Inequality and Relative Index of Inequality further enhances this paper. Though this section would be strengthened with the inclusion of further information regarding the sampling and design weights.
· 3. Figures 1 and 2 are quite difficult to interpret as it is not always possible to see the bars for 2013 and 2019 across each region, as they are placed on top of each other.
Author Response
Reviewer #2
Reviewer #2: Thank you for the opportunity to review this manuscript. The authors present a cross-sectional analysis of data from the 2013 and 2019 National Health Survey, with large and representative samples, which explores socioeconomic inequalities in self-rated oral health. The introduction provides a comprehensive overview of existing literature and highlights the ways in which socioeconomic factors contribute to oral health, particularly in context of the National Oral Health Policy that was introduced in Brazil in 2004 and sought to reduce inequalities in oral health. The authors find that individuals with lower socioeconomic indicators had a lower prevalence of positive self-rated oral health in both 2013 and 2019, highlighting the persistence of the social gradient for self-rated oral health. Further, they show that a reduction in the magnitude or relative education-based and income-based inequalities have been observed in some regions in Brazil and this is associated with positive self-rated oral health. The findings have important implications for equitable policies which seek to reduce self-rated oral health inequalities. This is a very well-written manuscript, with appropriate statistical analyses, and novel findings. I have only minor suggestions for revisions.
Author response: Thank you for your constructive comments.
Reviewer #2 – point 1:
In the methods section, it could be clearer whether the National Health Survey is a repeated cross-sectional study, i.e., did the study ask the same questions in 2013 and 2019? And were different individuals recruited for each study?
Author response: We appreciated your comment. To clarify this information, we have added a sentence in the first paragraph of the Materials and Methods section to indicate that the two surveys were conducted using similar methodologies but involved different populations. The modules and questions used in the PNS were the same in both years, allowing data analysis in the period. This information is presented in the third paragraph of the Materials and Methods section.
This analytical cross-sectional study used public secondary data from two national health surveys (NHS) conducted in Brazil in 2013 and 2019. These two surveys were carried out using similar methodologies but involved different populations, resulting in independent samples for each year. The NHS is a household health survey developed with the scope of health surveillance and assistance in partnership with the Ministry of Health, the Oswaldo Cruz Foundation, and the Brazilian Institute of Geography and Statistics (IBGE) [11]. The databases and variables dictionaries for 2013 and 2019 were obtained from the IBGE website in June 2022. The database version contained updates and corrections made to the 2013 (updated on 08/25/2020) and 2019 (updated on 25/05/2022). These included corrections for sample weight based on the Population Projection of the Federation Units by sex and age for 2010 – 2060.
Reviewer #2 – point 2:
The statistical analyses are described in great detail and the use of the Slope Index of Inequality and Relative Index of Inequality further enhances this paper. Though this section would be strengthened with the inclusion of further information regarding the sampling and design weights.
Author response: Thank you for your suggestion. We have added information about the calculation of sample weights to the second paragraph of the Materials and Methods section.
In each year of the NHS, the sample was selected from residents residing in permanent private households across Brazilian urban and rural areas, encompassing five geographic macro-regions, federative units, capitals, and metropolitan regions [12]. The NHS sample is a subsample of the IBGE Master Sample, used as unities of many areas selected to be used in several national surveys. To determine the sample size required for estimating parameters of interest across various levels of geographic disaggregation in the NHS, several factors were taken into account: the estimated proportions and the desired level of precision within 95% confidence intervals (95% CI), the design effect (Deff) due to the multi-stage cluster sampling method used, the number of households selected per Primary Sampling Unit (PSU) and the proportion of households containing individuals within the target age group [13]. The sample selection occurred in three stages using a simple random draw: census sectors or sets of sectors (Primary Sampling Units-PSU), permanent private households (Second Sampling Stage), and adults residing in these households (Third Sampling Stage). For each PSU, 10 to 14 households were randomly selected, depending on the domain size, to reach the minimum sample required. Within each household, one resident was chosen randomly with equal probability among eligible participants. Sampling weights were defined for the primary sampling units, households, and all their residents. Further details on the sampling procedure and weighting factors can be found in previous publications [12, 14]. The ages of interest were over 18 years in the 2013 NHS [14] and over 15 years in the 2019 NHS [11]. For this study, data for adults under 18 years from the NHS 2019 [14] were excluded to allow comparison between surveys [10].
Additionally, we included the following details about the analyses performed: All analyses were performed using the Stata statistical package, version 18.0 (StataCorp LP, College Station, TX, USA), accounting for complex survey design and sampling weights employed using the “svy” command.
We used generalized linear models (log-binomial regression) with an identity link function to calculate the SII (rate differences) and a logarithmic link function to calculate the RII (rate ratios). Both indices were estimated with 95% confidence intervals and were adjusted for sex and age. Statistical significance was set at P < 0.05. All analyses were performed using the Stata statistical package, version 18.0 (StataCorp LP, College Station, TX, USA), accounting for complex survey design and sampling weights employed using the “svy” command.
Reviewer #2 – point 3:
Figures 1 and 2 are quite difficult to interpret as it is not always possible to see the bars for 2013 and 2019 across each region, as they are placed on top of each other.
Author response: Thank you for your suggestion. In order to help the interpretation, we changed the graphic presentation as showed below.

Figure 1. Comparison of prevalence of positive SROH according to educational levels in Brazilian regions in 2013 and 2019.

Figure 2. Comparison of prevalence of positive SROH according to income levels in Brazilian regions in 2013 and 2019.

Reviewer 3 Report
Comments and Suggestions for Authors
I have read with special interest this manuscript. The topic is very interesting for the journal readers from a public health perspective in the dentistry field. The manuscript is well-written.
I have a list of few recommendations:
- Please provide the STROBE checklist, in order to verify the accomplishment of all requirements for observational studies (cross-sectional).
- Lately, Most Journals, recommend to conduct gender analyses (separately for Men and Women, that it means, the sex/gender is not just an additional variable). Since the authors use the variable sex, please provide a justification or discuss the importance to study this perspective (study limitation).
- Please provide recommendations for research and practice as derived from the study findings
Author Response
Reviewer #3: I have read with special interest this manuscript. The topic is very interesting for the journal readers from a public health perspective in the dentistry field. The manuscript is well-written.
Author response: Thank you for your feedback. We are pleased to hear that you found our manuscript interesting.
Reviewer #3 – point #1: Please provide the STROBE checklist, in order to verify the accomplishment of all requirements for observational studies (cross-sectional).
Author response: Thank you for your suggestion. Including the STROBE Checklist allowed us to thoroughly review the manuscript and incorporate additional details to enhance clarity and completeness. We have now attached the completed STROBE Checklist for your reference.STROBE Statement
—Checklist of items that should be included in reports of cross-sectional studies
|
|
Item No |
Recommendation |
|
Title and abstract |
1 |
(a) Indicate the study’s design with a commonly used term in the title or the abstract This was a cross-sectional study as stated in the opening words of the Abstract presented on page 1. |
|
(b) Provide in the abstract an informative and balanced summary of what was done and what was found The abstract includes the objetives, materials and methods, main results and conclusions. . |
||
|
Introduction |
||
|
Background/rationale |
2 |
Explain the scientific background and rationale for the investigation being reported Introduction (1st to 3rd paragraphs) |
|
Objectives |
3 |
State specific objectives, including any prespecified hypotheses Last paragraph of Introduction. |
|
Methods |
||
|
Study design |
4 |
Present key elements of study design early in the paper 1st paragraph: we described the study design. |
|
Setting |
5 |
Describe the setting, locations, and relevant dates, including periods of recruitment, exposure, follow-up, and data collection 1st paragraph. |
|
Participants |
6 |
(a) Give the eligibility criteria, and the sources and methods of selection of participants The sample selection was described on the second paragraph of Material and Methods, presented on pages 2 and 3. |
|
Variables |
7 |
Clearly define all outcomes, exposures, predictors, potential confounders, and effect modifiers. Give diagnostic criteria, if applicable The outcome was described in the 4th paragraph. Socioconomic indicators and covariates was presented in the 5th and 6th paragraphs. |
|
Data sources/ measurement |
8* |
For each variable of interest, give sources of data and details of methods of assessment (measurement). Describe comparability of assessment methods if there is more than one group The variables, categories and measurement of the outcome and exposures were described in the fourth, fifth and sixth paragraphs of Material and Methods. |
|
Bias |
9 |
Describe any efforts to address potential sources of bias We described the methods to collecting data and the analytical approach to adjust the regression model. |
|
Study size |
10 |
Explain how the study size was arrived at The sample size of study was calculated to estimating parameters of interest considering : the estimated proportions and the desired level of precision within 95% confidence intervals (95% CI), the design effect (Deff) due to the multi-stage cluster sampling method used, the number of households selected per Primary Sampling Unit (PSU) and the proportion of households containing individuals within the target age group.. This information was added in the 2nd paragraph of the Material and Methods section. |
|
Quantitative variables |
11 |
Explain how quantitative variables were handled in the analyses. If applicable, describe which groupings were chosen and why Included on page 3 of the Material and Methods section. |
|
Statistical methods |
12 |
(a) Describe all statistical methods, including those used to control for confounding Included in the Material and Methods, pages 4 and 5. |
|
(b) Describe any methods used to examine subgroups and interactions Included in the Material and Methods, pages 4 and 5. |
||
|
(c) Explain how missing data were addressed Included in the Material and Methods, pages 4 and 5. |
||
|
(d) If applicable, describe analytical methods taking account of sampling strategy Included in the Material and Methods, pages 4 and 5. |
||
|
(e) Describe any sensitivity analyses We performed logistic regression models and estimated the magnitude of disparities. Both approaches are consistent in their findings regarding differences in outcome levels relative to the analyzed socioeconomic indicators. |
||
|
Results |
||
|
Participants |
13* |
(a) Report numbers of individuals at each stage of study—eg numbers potentially eligible, examined for eligibility, confirmed eligible, included in the study, completing follow-up, and analysed We described the participants included in first paragraph of results. |
|
(b) Give reasons for non-participation at each stage The non-response rates for some variables are detailed in the first paragraph. |
||
|
(c) Consider use of a flow diagram A flow diagram was deemed unnecessary due to the very low amount of missing data. |
||
|
Descriptive data |
14* |
(a) Give characteristics of study participants (eg demographic, clinical, social) and information on exposures and potential confounders Presented on page 4 and 5 in the Results section (Table 1). |
|
(b) Indicate number of participants with missing data for each variable of interest Included on first paragraph of Results on page 4. |
||
|
Outcome data |
15* |
Report numbers of outcome events or summary measures Included on Results on pages 4 and 5. |
|
Main results |
16 |
(a) Give unadjusted estimates and, if applicable, confounder-adjusted estimates and their precision (eg, 95% confidence interval). Make clear which confounders were adjusted for and why they were included Included in the Results section, pages 4 to 8, and summarized in the tables. |
|
(b) Report category boundaries when continuous variables were categorized Included in the Results section, pages 4 to 8, and summarized in the tables |
||
|
(c) If relevant, consider translating estimates of relative risk into absolute risk for a meaningful time period Not applicable. |
||
|
Other analyses |
17 |
Report other analyses done—eg analyses of subgroups and interactions, and sensitivity analyses The Materials and Methods section and supplementary file presented the regression model and the results of the SII and RII, taking into account the evaluated socioeconomic indicators. |
|
Discussion |
||
|
Key results |
18 |
Summarise key results with reference to study objectives Included on the Discussion section, on its first paragraphs presented on pages 7 and 8. |
|
Limitations |
19 |
Discuss limitations of the study, taking into account sources of potential bias or imprecision. Discuss both direction and magnitude of any potential bias Included on the last paragraph of Discusson, on page 9. |
|
Interpretation |
20 |
Give a cautious overall interpretation of results considering objectives, limitations, multiplicity of analyses, results from similar studies, and other relevant evidence Included in the Discussion on pages 7, 8 and 9. |
|
Generalisability |
21 |
Discuss the generalisability (external validity) of the study results Provided in the last paragraph of Discussion. |
|
Other information |
||
|
Funding |
22 |
Give the source of funding and the role of the funders for the present study and, if applicable, for the original study on which the present article is based Indicated on page 13. |
Reviewer #3 – point #2: Lately, Most Journals, recommend to conduct gender analyses (separately for Men and Women, that it means, the sex/gender is not just an additional variable). Since the authors use the variable sex, please provide a justification or discuss the importance to study this perspective (study limitation).
Author response: Thank you for your suggestion. In this study, we do not aimed to analyze the results according to the sex of the participants. Then, we included this information in the study limitation as highlighted in red words.
The strengths of this study are that the data were obtained from nationally representative health surveys and included education and income levels, which are the most common proxies of social position for measuring absolute and relative inequalities. Given the sample calculation, the findings are representative of the population of Brazil as a whole and of each Brazilian region. To the best of our knowledge, this is the first study to evaluate changes in the magnitude of socioeconomic inequalities of SROH among Brazilian people using a two-way interaction term ridit score to analyze modification in SII and RII over time. The design of the NHS excludes the homeless population and residents of long-stay institutions. In addition, the interviews were carried out with only one resident. While it is acknowledged that conducting a gender analysis is important because men and women may have different risk factors, access to care, biological influences, and social determinants that impact their oral health outcomes, this study opted to adjust the estimates only by sex. Future research could stratify by sex to ensure a comprehensive understanding of oral health disparities and promote more equitable healthcare for all. Some measurement bias may have occurred because socioeconomic indicators were self-reported. (…)
Reviewer #3 – point #3: Please provide recommendations for research and practice as derived from the study findings.
Author response: Thank you for your consideration. We included the recommendations for research and practice as showed below, as also suggested by the first reviewer.
The strengths of this study are that the data were obtained from nationally representative health surveys and included education and income levels, which are the most common proxies of social position for measuring absolute and relative inequalities. Given the sample calculation, the findings are representative of the population of Brazil as a whole and of each Brazilian region. To the best of our knowledge, this is the first study to evaluate changes in the magnitude of socioeconomic inequalities of SROH among Brazilian people using a two-way interaction term ridit score to analyze modification in SII and RII over time. The design of the NHS excludes the homeless population and residents of long-stay institutions. In addition, the interviews were carried out with only one resident. While it is acknowledged that conducting a gender analysis is important because men and women may have different risk factors, access to care, biological influences, and social determinants that impact their oral health outcomes, this study opted to adjust the estimates only by sex. Future research could stratify by sex to ensure a comprehensive understanding of oral health disparities and promote more equitable healthcare for all. Some measurement bias may have occurred because socioeconomic indicators were self-reported. The subjectivity embedded in the evaluation of SROH is influenced by circumstances in a person’s life, day, and week and is a result of the contextual and psychosocial conditions experienced by the individual, involving values and feelings that are not expressed. This characteristic provides a comprehensive assessment of how people perceive their oral health and its effects of oral health on the functional, social, and psychosocial aspects of daily life. Subjectivity qualifies the relevance of the findings to health policies and decision-making. Despite our limitations, our results represent the inequalities in SROH regarding income and education in Brazil. Research and policies that focus on a more equitable distribution of power, prestige, opportunities, and resources in income and education could improve health conditions and alleviate the negative perceptions of oral health among marginalized individuals.

Round 2
Reviewer 1 Report
Comments and Suggestions for Authors
The revised version is acceptable.